# Analytical Study of the Impact of Solidity, Chord Length, Number of Blades, Aspect Ratio and Airfoil Type on H-Rotor Darrieus Wind Turbine Performance at Low Reynolds Number

Pedram Ghiasi [1], Gholamhassan Najafi [1,*], Barat Ghobadian [1], Ali Jafari [2] and Mohamed Mazlan [3,*]

1  Department of Biosystems Engineering, Tarbiat Modares University, Tehran P.O. Box 111-14115, Iran; p.ghiasi@modares.ac.ir (P.G.); ghobadib@modares.ac.ir (B.G.)
2  Department of Agricultural Engineering, University of Tehran, Karaj P.O. Box 6619-14155, Iran; a.jafary@ut.ac.ir
3  Advanced Material Cluster, Faculty of Bioengineering and Technology, University Malaysia Kelantan, Jeli, Locked Bag No. 100, Kelantan 17600, Malaysia
*  Correspondence: g.najafi@modares.ac.ir (G.N.); mazlan.m@umk.edu.my (M.M.)

**Abstract:** The use of wind energy can be traced back thousands of years to many ancient times. Among the tools used for converting wind energy was the vertical-axis wind turbine (vawt). Investigating the performance of this type of turbine is an interesting topic for researchers. The appropriate correlation between the Double Multiple Stream Tube (DMST) model and the experimental results has led researchers to pay distinct attention to this model for vawt simulation. In this study, using the aforementioned model, the appropriate range of important wind turbine design parameters was determined. First, the model outcome was validated based on experimental results and then, the performances of 144 different turbine types were simulated with respect to chord length, number of blades, H/D ratio and airfoil type. Chord length was evaluated at three levels, 0.1, 0.15 and 0.2 m, number of blades 2, 3 and 4, Height to Diameters (H/D) ratio of 0.5, 1, 1.5 and 2, and four types of airfoils, NACA0012, NACA0018, NACA4412 and NACA4418. Simulation was performed at a low Reynolds number (Re $\leq 10^5$) and at four TSRs, 1, 2, 3 and 4. The results show that wind turbines perform best at low TSRs when they have longer chords, more blades, and a higher H/D ratio, but this trend reverses at high TSRs. Among the four types of airfoils evaluated, the NACA4412 airfoils showed a better performance at TSRs 1 to 3.

**Keywords:** DMST model; angle of attack; power coefficient

## 1. Introduction

In recent years, wind turbines have been considered a reliable tool in the field of power supply [1]. Vertical-axis wind turbines (VAWT) have advantages over horizontal-axis turbines due to their unique features [2]. This type of turbine is notable for its insensitivity to wind direction, low construction costs, low installation and maintenance costs, high adaptability, and lower acoustic noise due to its lower blade tip operation compared to horizontal axis wind turbines (HAWT) [3]. HAWTs rotate by lift force but the VAWTs can be operated by lift or drag force [4]. In general, the speed of the turbine blade tip in a drag regime (such as Savonius) does not exceed one, but they can rotate at low wind speeds [5]. Nevertheless, Darrieus vertical axis wind turbines function properly when the blade tip speed is above 1, and when it enters the lift regime [6]. One of the problems of these turbines is the low initial torque at a low Reynolds number. In general, H-rotor Darrius wind turbines hardly pass the drag regime and enter the lift mode. When the Reynolds number of wind is low, this problem culminates. Manipulating the design parameters to overcome these problems is of great interest in this field and it requires each researcher to provide a suitable range for the design parameters of Darrieus wind turbines.

Generally, the use of different airfoils can change the power and momentum coefficients. Of the different type of airfouls, the EN0005 takes the least amount of time to achieve a stable rotational speed. Yet, taking into account other parameters such as power coefficient and torque stability, the airfoil S-1046 bears the best performance for a wind turbine. This comes from numerical studies on the Darrieus wind turbine with straight blades [7]. Other findings confirm the fact that the rotation at low Reynolds is more likely achievable for curved airfoils than for symmetrical ones [8–10]. Dynamic stall typically occurs more at lower blade tip speeds due to the increased angle of attack. There are hardly any research reports demonstrating the phenomenon; this is why Tirandaz and Rezaeiha [11] aimed at studying the effect of airfoil shape on a dynamic stall.

Their work on 126 types of airfoils with different shapes showed that increasing the leading-edge radius kindles a decrease in power coefficient in all cases, and increasing the blade thickness at low blade speeds can improve turbine performance, but this improvement was not observed at high blade speeds [12]. In virtue of increasing the number of blades, the problem of wind turbines' self-starting might be overcome to some extent, but escalating the number of blades disrupts the rotor operation at high blade tip speeds [13]. Moreover, by and large, rotors with high solidity may reach a stable rotational speed faster but perform poorer at lower blade tip speeds [14]. One of the valid models being hired for simulating wind turbine performance in many studies is the DMST model [15–17]. The DMST model was used to select the suitable airfoil for utilization in a Darrieus wind turbine and it was found that NACA0018 airfoil provides better performance [18]. Although higher blade solidity increases the power coefficient at lower blade speeds, it greatly declines the turbine power coefficient at high blade speeds. This result was obtained from the simulation of Darrieus wind turbine with theDMST model [19].

Many studies have been undertaken on high Reynolds and high TSRs for Darrieus turbines, but the biggest problems with this type of turbine occurred regarding low Reynolds and low blade tip speeds. As a whole, investigations on low Reynolds and TSR<5 also include limited parameters. On this point, exploring various parameters in large ranges may play a vital role in understanding the performance of this type of turbine and also tackling the relevant challenges. The aim of the current study is to determine the appropriate range for designing parameters such as chord length, number of blades, H/D ratio and airfoil type to achieve a better performance of Darrieus vertical-axis wind turbines at low Reynolds. Since the performance of wind turbines is different at each TSRs, in each TSR, the appropriate designing parameters were defined. This study is a guide for designing Darrieus turbines at a low Reynolds to overcome self-starting problems.

## 2. Materials and Methods

### 2.1. Turbines

In this study, 144 types of turbines with different aspect ratios, chord lengths, number of blades and different types of airfoils were investigated. The aspect ratio of the rotor runs from 0.5 to 2 with a step of 0.5. Within this range, the aspect ratio of the rotor with two, three, and four blades was set. Figure 1 shows 12 wind turbines with different aspect ratios and numbers of blades. Other parameters exerting influence on the performance of wind turbines are airfoil type and its chord length. Airfoils are classified based on thickness, shape and amount of curvature. Based on the previous studies the NACA 4-digit series is commonly used in the H-rotor Darrius wind turbine [11,20], so in this study four types of airfoils were selected from the NACA 4-digit series with two thicknesses (symmetrical and asymmetrical). NACA4418, NACA4412, NACA0018 and NACA0012 is a list of four airfoils used in turbines. Figure 2 demonstrates the cross-sectional view of each of three chord lengths, 0.1, 0.15 and 0.2 m. The turbines were designed so that the area in front of them was approximately 2 $m^2$. The turbines' height ranged from 1 to 2 m, and their radius ranged from 0.5 to 1 m, with different combinations of these sizes resulting in different aspect ratios for the turbines. Because the goal of this study is to determine the design parameters to overcome the problems associated with power generation in wind turbines

at low Reynolds numbers, all numerical simulations were performed at 6 m·s$^{-1}$ of free stream velocity and low Reynolds numbers ranging from 50,000 to 100,000. Tip Speed Ratio (TSR) was calculated from r$\omega$/V$_\infty$ ratio and the performances of turbine in 1, 2, 3 and 4 of TSRs were investigated.

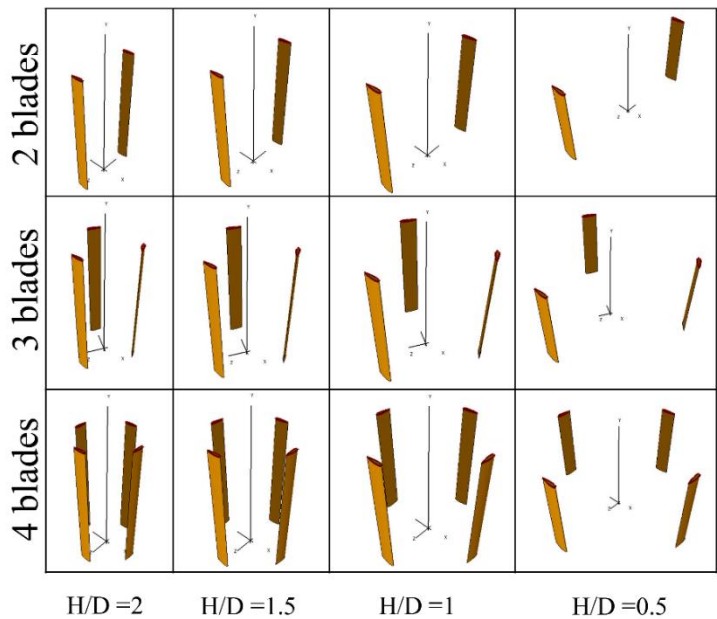

**Figure 1.** 12 deffirent rotor with varius number of blades and H/D ratio.

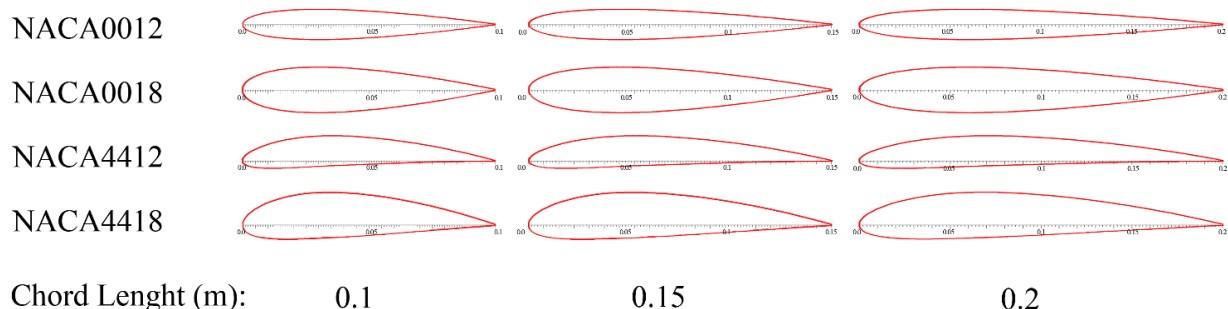

**Figure 2.** 4 airfoil shape with three chord length.

### 2.2. Double Multiple Stream Tube (DMST) Model

In the present study, A DMST model, which was originally developed by (Paraschivoiu, 1988), was utilized [20]. The model is a hybrid of the MST model and the theory of double actuators. Upwind and downwind sections of turbines were separated. At a high Reynolds number and high rotor solidity, the correlation between model results and experimental results decreases. One of the strengths of this study is the use of the DMST model in the low Reynolds number. The first step was to calculate the forces acting on the VAWT blades, which required the value and direction of velocities to be calculated. The upstream and downstream velocities differed from the free velocity of the wind (Figure 3). The velocity region between up and down streams is represented by $V_e$. The following are the Equations [21]:

$$V = uV_\infty \tag{1}$$

$$V_e = (2u - 1)V_\infty \tag{2}$$

$$V' = u'(2u - 1)V_\infty \tag{3}$$

where $V_\infty$ is the free stream wind velocity, $V$ is upstream wind velocity, $V_e$ is equilibrium velocity and $V'$ is the velocity of wind in downstream section.

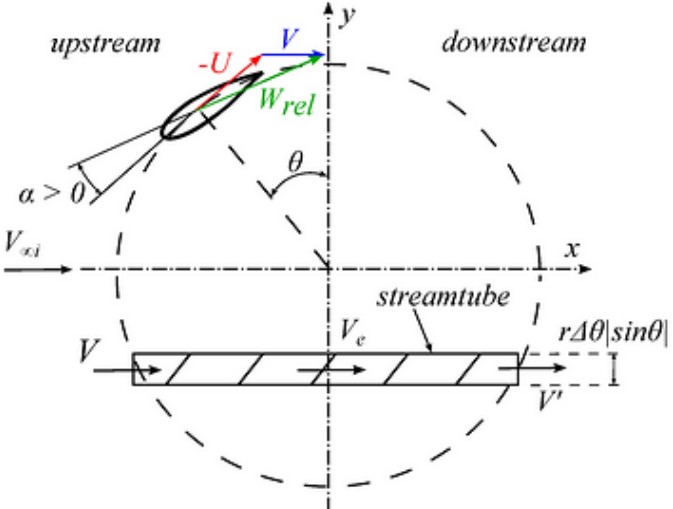

**Figure 3.** Principle of DMST model [18].

The upstream section of the rotor's local relative wind speed (W) was determined by [19]

$$W = V_\infty \sqrt{(\lambda_0 + u sin(\theta))^2 + (u cos(\theta))^2} \tag{4}$$

The angle of attack was calculated by:

$$\alpha_u = \tan^{-1}\left(\frac{u cos(\theta)}{\lambda_0 + u sin(\theta)}\right) \tag{5}$$

The force on blades must be calculated in order to determine the induction factors derived from blade element theory [22]. The induction factors for each stream tube at the rotor can be calculated by combining blade element theory and momentum theory.

$$u = \frac{\pi}{f_u + \pi} \tag{6}$$

$$f_u = \frac{\sigma}{2\pi} \int_0^\pi \left(\frac{W}{V_\infty}\right)^2 |sec(\theta)| (C_n cos(\theta) - C_t sin(\theta)) d\theta \tag{7}$$

where the blade solidity is calculated as follows:

$$\sigma = \frac{Nc}{2R} \tag{8}$$

where $C_n$ and $C_t$ are the coefficients of the force's normal and tangential components, respectively. $C_n$ and $C_t$, as calculated by Equations (9) and (10), are functions of drag and lift coefficients as well as blade AOA:

$$C_n = C_L cos(\alpha) + C_D sin(\alpha) \tag{9}$$

$$C_t = C_L sin(\alpha) - C_D cos(\alpha) \tag{10}$$

The normal and tangential forces acting on blades can be calculated as follows:

$$F_n(\theta) = \frac{A_p}{A_s} C_n \left(\frac{W}{V_\infty}\right)^2 \tag{11}$$

$$F_t(\theta) = \frac{A_p}{A_s} C_t \left(\frac{W}{V_\infty}\right)^2 \tag{12}$$

The tangential force is used to calculate torque near the rotor's center, so

$$T_{up}(\theta) = \frac{\rho c R H C_t W^2}{2} \tag{13}$$

All of the equations are repeated for the downstream of the rotor, and the free wind velocity is replaced by the equilibrium velocity determined by Equation (2). The local relative wind speed, AOA, and modified AOA equations are defined as follows:

$$W' = (2u - 1)V_\infty \sqrt{\left(\frac{\lambda_0}{(2u-1)} + u'sin(\theta)\right)^2 + (u'\cos(\theta))^2} \tag{14}$$

$$\alpha_u' = \tan^{-1}\left(\frac{u'\cos(\theta)}{\frac{\lambda_0}{(2u-1)} + u'sin(\theta)}\right) \tag{15}$$

The induction factor in the downstream region of the rotor is determined similarly to that in the upstream region, but the value of the induction factor in the downstream region is less than that of the upstream region.

$$u' = \frac{\pi}{f_d + \pi} \tag{16}$$

$$f_d = \frac{\sigma}{2\pi} \int_\pi^{2\pi} \left(\frac{W}{V_\infty}\right)^2 |\sec(\theta)| (C_n'\cos(\theta) - C_t'\sin(\theta)) d\theta \tag{17}$$

After calculating the induction factor for the downstream region, the coefficients of the normal and tangential components of the force, *Cn'* and *Ct'*, are obtained from Equations (19) and (20), respectively.

$$C_n' = C_L\cos(\alpha') + C_D\sin(\alpha') \tag{18}$$

$$C_t' = C_L\sin(\alpha') - C_D\cos(\alpha') \tag{19}$$

Equations (20) and (21) determine the components of the force acting on the blades downstream.

$$F_n(\theta) = \frac{A_p}{A_s} C_n' \left(\frac{W'}{V_\infty}\right)^2 \tag{20}$$

$$F_t(\theta) = \frac{A_p}{A_s} C_t' \left(\frac{W'}{V_\infty}\right)^2 \tag{21}$$

The torque and power coefficients generated in the downstream region are expressed as follows:

$$T_{down}(\theta) = \frac{\rho c R H C_t' W'^2}{2} \tag{22}$$

$$\overline{C}_{P_{down}} = \frac{\sigma}{4\pi} \int_\pi^{2\pi} C_t' \left(\frac{W'}{V_\infty}\right)^2 d\theta \tag{23}$$

$$C_{P_{down}} = \lambda_0 . \overline{C}_{P_{down}} \tag{24}$$

The total power coefficient of the rotor is calculated by adding the power coefficients of each region [23], which is as follows:

$$C_p = C_{P_{up}} + C_{P_{down}} \tag{25}$$

Finally, the momentum coefficient was determined as follow:

$$C_m = \frac{M}{\frac{1}{2}\rho ARV_\infty^2} \tag{26}$$

All parameters were defined in the Appendix A Table A1.

### 2.3. Model Validation

Validated data were used to verify the DMST model solutions. Experimental results of the study of Darrieus wind turbine were reported by Elkhoury et al. (2015) [24]. They numerically and experimentally investigated the performance of a micro scale H type three bladed Darrieus VAWT with high solidity σ = 0.75, length of 0.8 m and diameters of 0.8 m. It had a symmetrical blade profile (NACA 0018) with a chord length of 0.2 m. A power coefficient at 0–3 TSR with three wind velocities of 6, 8 and 10 m·s$^{-1}$ was used for verifying the process. Figure 4 shows the comparison between experimental data of Cp in 8 m·s$^{-1}$ of wind velocity and DMST model simulation. The results showed a relatively good match of estimated with experimental data. Assuming the constant amount of velocity, frontal area and air density the relationship between power coefficient and TSR is the same. The maximum difference between the experimental data and DMST data belongs to TSR 1 with amount 8%.

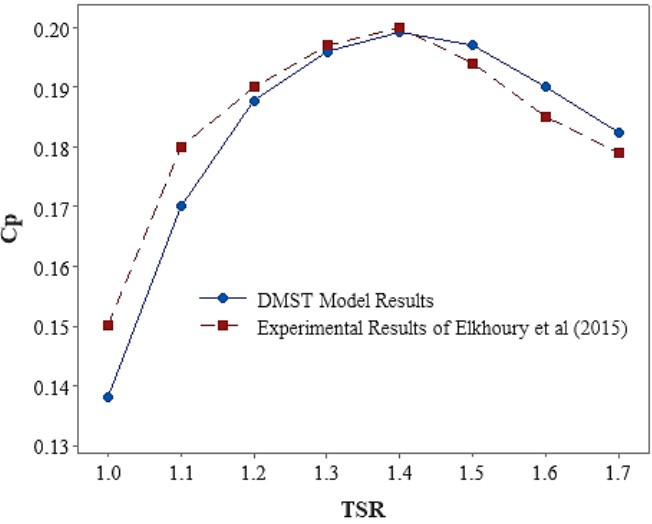

**Figure 4.** Experimental and DMST model results.

### 3. Results

#### 3.1. Design Parameters

Power and momentum coefficients reflect the capability of a turbine to transform wind energy. Figures 5–8 exhibit the power and momentum output coefficients of the DMST model for different types of wind turbines with blade NACA0012, NACA0018, NACA4412 and NACA4418 in different chord lengths, number of blades and H/D ratio. Sections (a) to (c) represent the power coefficient at TSRs from 1 to 4. At high TSRs, a reduction in chord length ameliorates turbine performance, but short-chord turbine power generation is disrupted at low TSRs. A change in chord length of the blade will alter the Reynolds number of flow around the blade and distort the lift-to-drag ratio of the blade. At an angle of attack of 5 to 45 degrees, increasing the Reynolds number will augment the NACA airfoil lift coefficient [25]. The secondary effect that the growing chord length will pose on the rotor is to change the solidity of the rotor, which increasing the solidity of the blade improves performance at low TSRs and diminishes performance at high TSRs. The momentum coefficient in sections (d) to (f) is shown in Figures 5–8. Concerning blades with high chord length, the increase in blade lift coefficient occurs at TSRs 1 and 2, leads

to increment in the momentum coefficient. This is due to the higher Reynolds number of blades with high chord length compared to turbines with smaller chord length. Increasing the number of blades strengthens the solidity of the rotor. Therefore, at low TSRs, an increase in power and momentum coefficient occurs for the rotor with more blades.

This applies for all four types of simulated airfoils, but the amount of increase in power and momentum coefficients in each of the airfoils is different, which will be examined in the following i.e., the effect of airfoil type on wind turbine performance. Figures 5 and 6 (section (b) and (e)) show that the maximum values of power and momentum coefficients are different in the three types of turbines, and turbines with fewer blades will perform better at higher TSRs. Altering the number of blades for turbines with NACA4412 and NACA4418 airfoil will cause different changes compared to symmetrical airfoils; so that the average power and momentum coefficient in the number of blades of less asymmetric airfoils at blade tip speeds 1 and 2 is likely more than turbines with symmetrical airfoils (section (b) and (e) of Figures 5–8). This may be effective for using and designing turbines in different situations. In general, with increasing blade solidity and blade asymmetry, the maximum power coefficient occurs at lower TSRs. These findings are obviously available in other studies [26,27]. Changes in the H/D ratio will change the amount of force created in each blade and the torque around the turbine shaft. On the other hand, the solidity of the rotor will also change by manipulating this parameter. Accordingly, H/D changes can affect turbine performance. Reducing the H/D ratio in the same frontal area will result in less solidity. Therefore, solidity is increased at higher H/D and the turbine is able to produce more power at lower TSRs. Jafari et al. [28] discovered a similar result for the effect of solidity on the maximum power coefficient. Figures 5–8, (section (c) and (f)) show the power and momentum coefficients of wind turbines with different H/D ratios. In NACA0012 and NACA4418 airfoil type turbines, the maximum momentum coefficient occurred for H/D of 1 ratio, but in NACA0018 and NACA4412 airfoil type turbines, the maximum power and momentum coefficients occurred for H/D of 0.5 ratio.

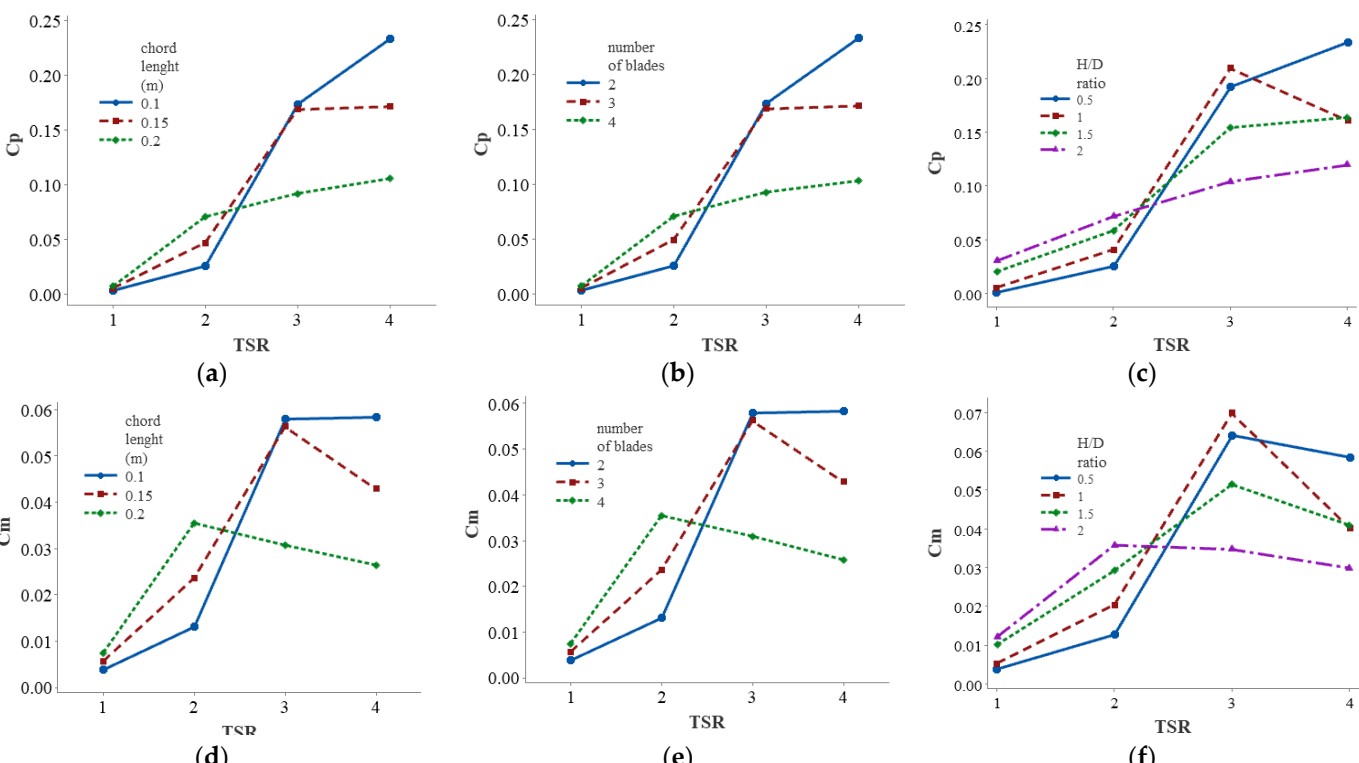

**Figure 5.** Power and momentum coefficient of rotor with NACA0012 airfoil. (**a**) effect of chord lenght on Cp. (**b**) effect of number of blade on Cp. (**c**) effect of H/D ratio on Cp. (**d**) effect of chord lenght on Cm. (**e**) effect of number of blade on Cm. (**f**) effect of H/D ratio on Cm.

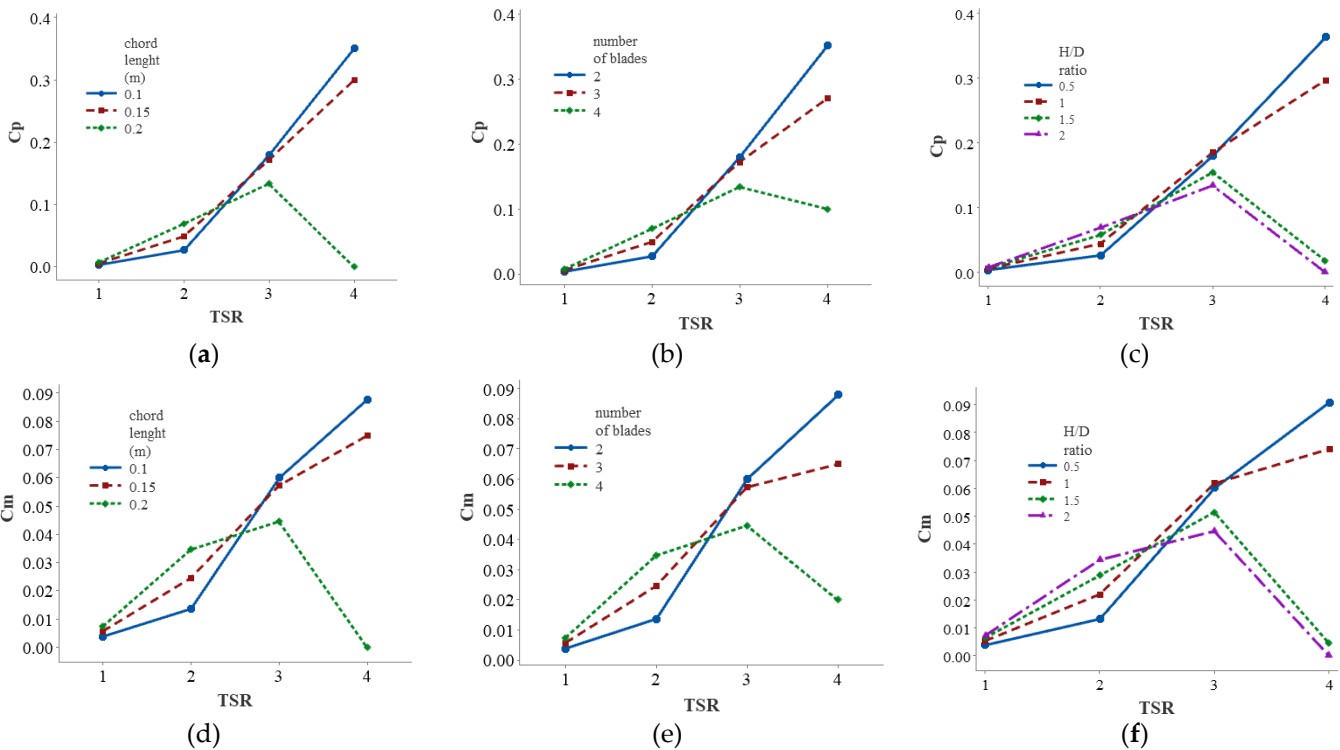

**Figure 6.** Power and momentum coefficient of rotor with NACA0018 airfoil. (**a**) effect of chord lenght on Cp. (**b**) effect of number of blade on Cp. (**c**) effect of H/D ratio on Cp. (**d**) effect of chord lenght on Cm. (**e**) effect of number of blade on Cm. (**f**): effect of H/D ratio on C.

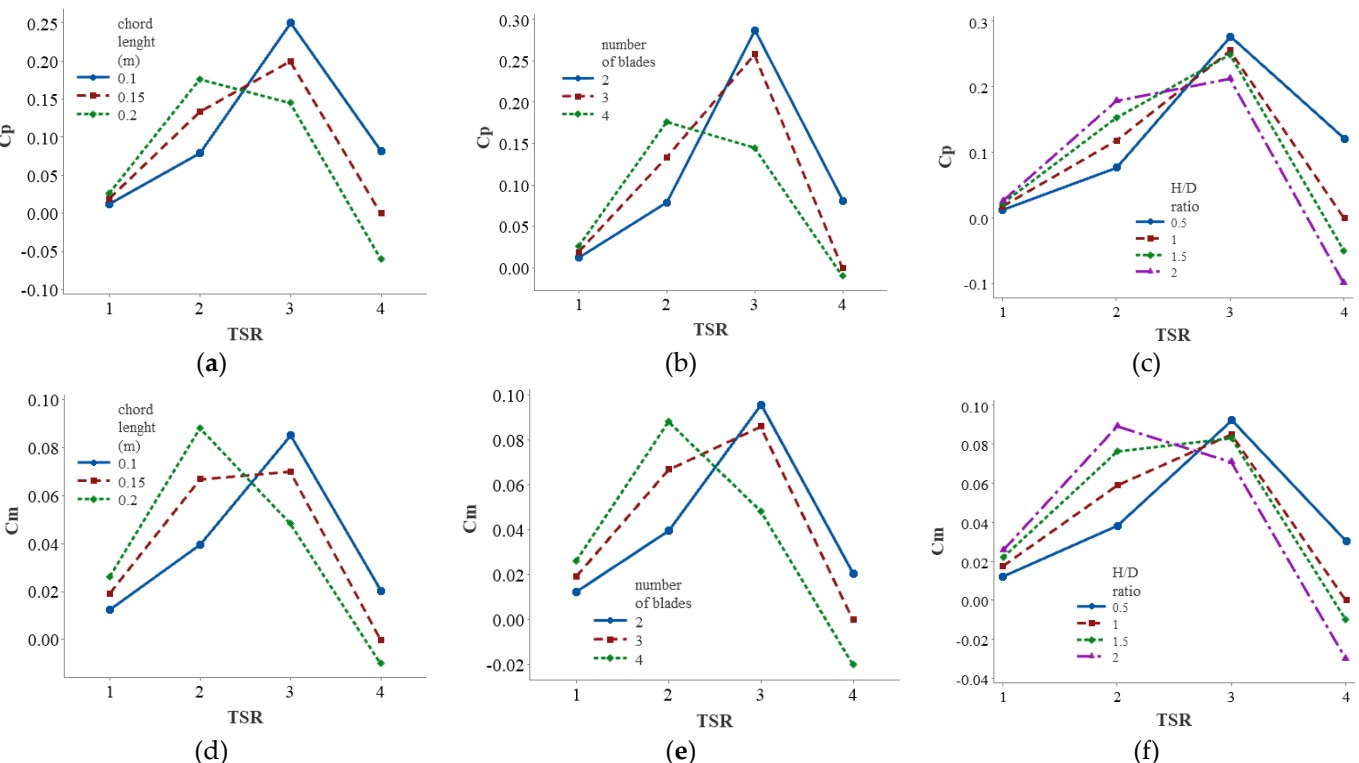

**Figure 7.** Power and momentum coefficient of rotor with NACA4412 airfoil. (**a**) effect of chord lenght on Cp. (**b**) effect of number of blade on Cp. (**c**) effect of H/D ratio on Cp. (**d**) effect of chord lenght on Cm. (**e**) effect of number of blade on Cm. (**f**) effect of H/D ratio on Cm.

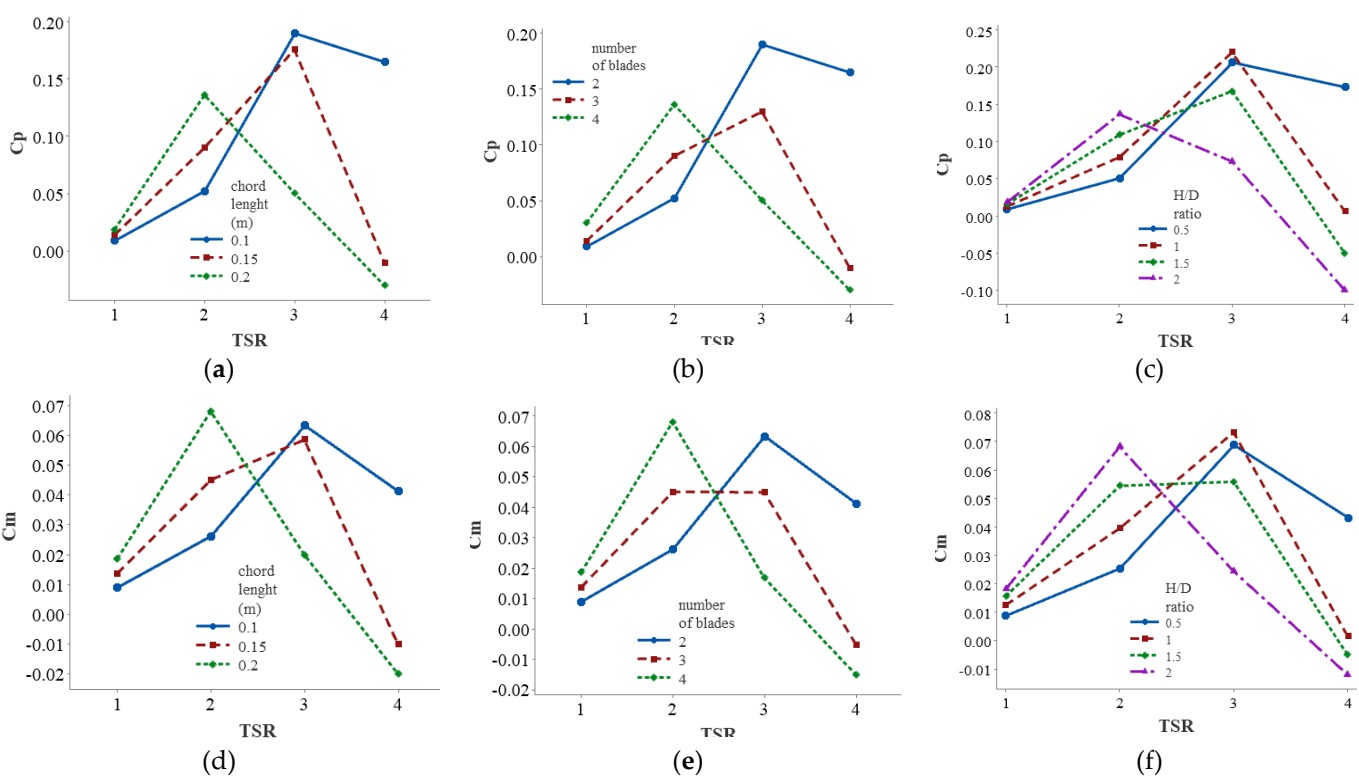

**Figure 8.** Power and momentum coefficient of rotor with NACA4418 airfoil. (**a**) effect of chord lenght on Cp. (**b**) effect of number of blade on Cp. (**c**) effect of H/D ratio on Cp. (**d**) effect of chord lenght on Cm. (**e**) effect of number of blade on Cm. (**f**) effect of H/D ratio on Cm.

Overall, Figures 4–8 present the similar data for four different airfoils. The similar behavior at different TSRs for the four types of airfoils indicates that to utilize the wind turbine in places with low average wind speeds, the use of asymmetric airfoils like NACA4412 can improve the performance of the turbine.

*3.2. Airfoil Type Effect*

The H/D ratio fashions a similar trend for turbines with symmetrical and asymmetrical airfoil types, but the order of power coefficient and momentum will be reversed at TSR 2.5 onwards. The turbine that has the highest power and momentum coefficients at TSR 1 and 2, will eventually be reduced to the minimum at TSR 3 and 4. The difference in the lift and drag coefficients of the airfoils accounts for the turbine's distinct performance with non-identical airfoils. Figure 9 depicts the average Cp and Cm of wind turbine for each TSR. To extract the data of Figure 9, the average of turbines with different chord length, H/D ratio and number of blades for four airfoil type were calculated. Increasing the thickness of the NACA0012 symmetrical airfoil from 12% to 18% at TSR 4 can almost double the power coefficient. Improved turbine performance also occurred with increasing symmetrical airfoil thickness from 12% to 18% at lower TSRs, but the most positive effect was related to TSR 4. The maximum power coefficient at TSR 4 occurred in both symmetrical airfoils, and at TSRs less than 3, and the coefficient of momentum and power was small. In NACA4412 and NACA4418 asymmetric airfoils, increasing the thickness resulted in a decrease in the turbine performance. The maximum momentum coefficient can be almost doubled by reducing the thickness of the asymmetric airfoil from 18% to 12%. The maximum power and momentum coefficients occurred in asymmetric airfoils at TSR 3 but then plummeted sharply. The maximum momentum coefficient was assigned to NACA4412 asymmetric airfoil and the maximum power coefficient to NACA0018 symmetric airfoil. By comparing asymmetric and asymmetric airfoils, conspicuously the power and momentum coefficients

of the turbine with symmetrical airfoils at TSR 4 are higher than asymmetric ones, but to overcome self-starting problems and increase momentum at low TSRs, it is preferably recommended using asymmetric airfoils.

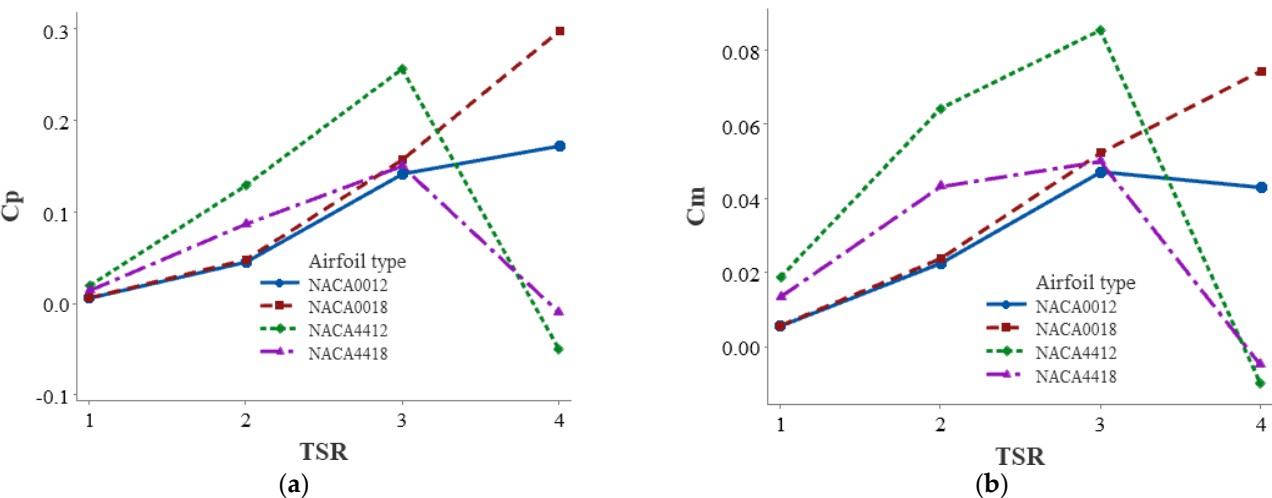

**Figure 9.** Effect of airfoil type on (**a**) Cp and (**b**) Cm.

### 3.3. Effects of Blade Solidity on Turbine Performance

The effects of solidity on the performance of Darrieus wind turbines have been investigated in previous sections. Since the effect of rotor solidity on power and momentum coefficients at different TSRs varies, for each TSR, power and momentum coefficients were displayed with respect to different rotor solidities. Figure 10a,b shows power coefficient changes of blade solidity in four types of symmetrical and asymmetrical airfoils. With increasing blade solidity to 0.8, the trend of power and momentum coefficients changes for all four types of airfoils increases, despite this, these values are negligible for symmetrical airfoils. The maximum power and momentum coefficients at TSR 1 occurred at rotor solidity of 0.8 for all four types of airfoils. NACA4412 airfoil was of significantly higher power and momentum coefficient values. A wealth body of experimental findings also confirms that the rotor will perform better at lower blade tip speeds with more solidity [29].

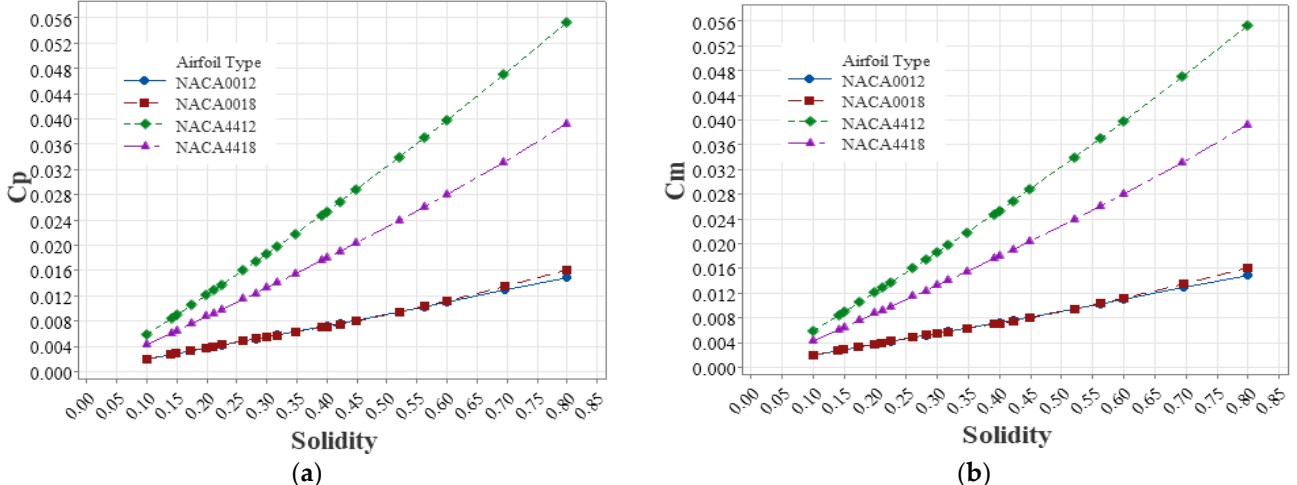

**Figure 10.** Effect of solidity on (**a**) Cp and (**b**) Cm in TSR = 1.

After NACA4412, NACA4418 airfoil had the most positive slope with respect to solidity along the increasing power and momentum coefficients. Figure 11a,b exhibits

the values of power and momentum coefficients at TSR 2 for the four types of airfoils at different solidity. At this TSR, the direct relationship between power and momentum coefficients with increasing solidity up to about 0.57 was continuous, but we observed a decrease in turbine performance with increasing solidity. The maximum point is nearly the same for all four types of turbines, but the NACA4412 airfoil performed best at this TSR. When differences are considered, the symmetry between the two symmetrical airfoils is similar up to solidity of 0.4, but then the NACA0012 airfoil increases and the NACA0018 airfoil exceeds it at solidity of 0.61; as at high solidity, symmetrical airfoil with higher thickness can work better. The power and momentum coefficients at TSR 3 vary and the solidity maximum points of the airfoils are in the range of 0.2 to 0.27 (Figure 12a,b). Figure 13a,b shows the power and momentum coefficients at TSR 4. In the high TSR the turbine with lower solidity can work better. Overall, the NACA4412 airfoil with solidity up to 0.39 had the best performance, however then dropped abruptly.

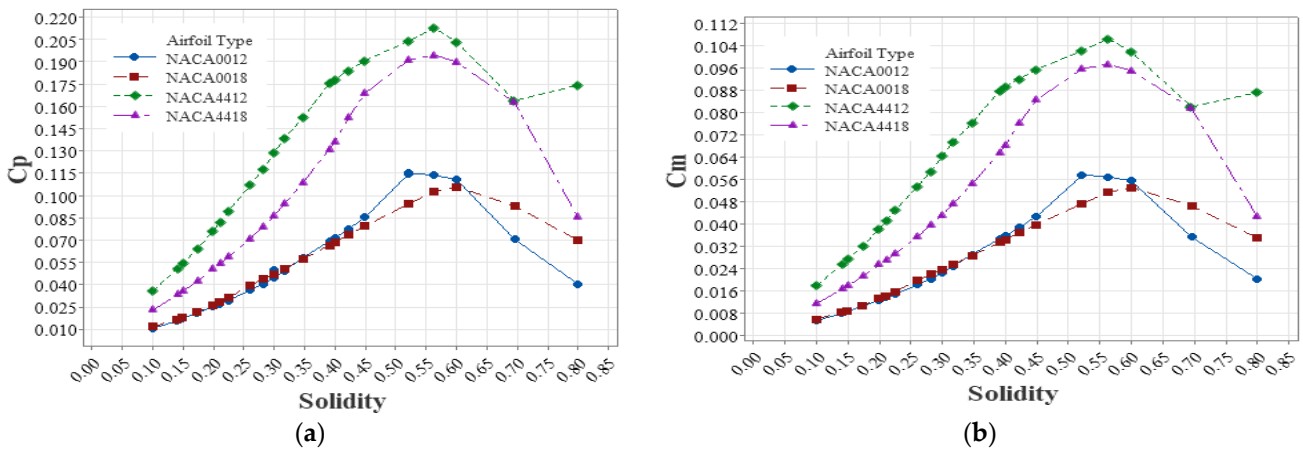

**Figure 11.** Effect of solidity on (**a**) Cp and (**b**) Cm in TSR = 2.

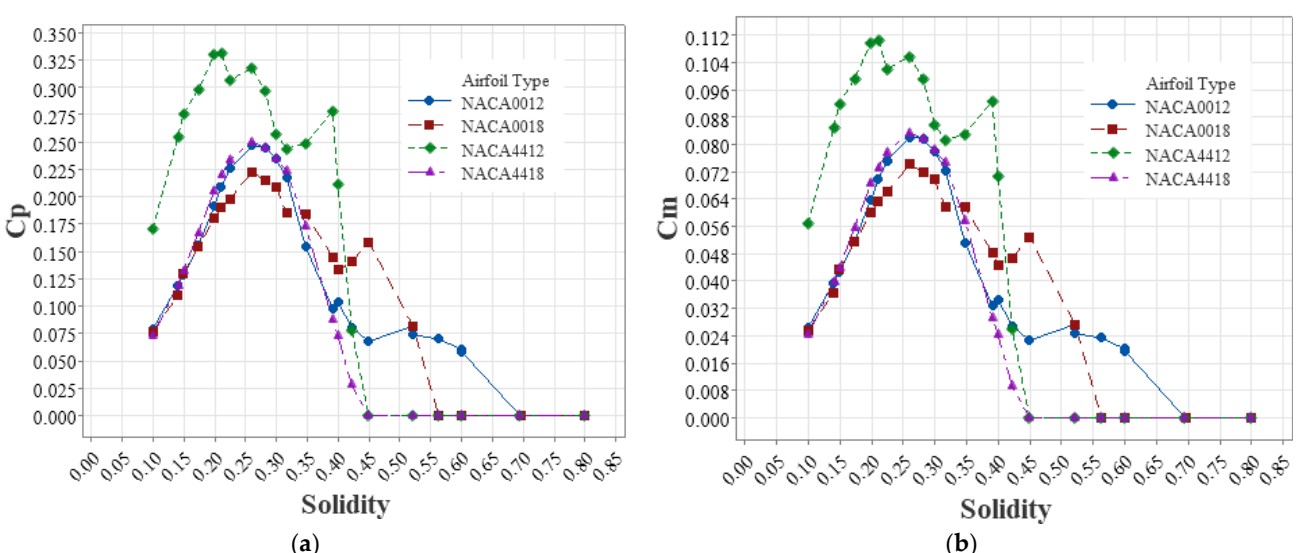

**Figure 12.** Effect of solidity on (**a**) Cp and (**b**) Cm in TSR = 3.

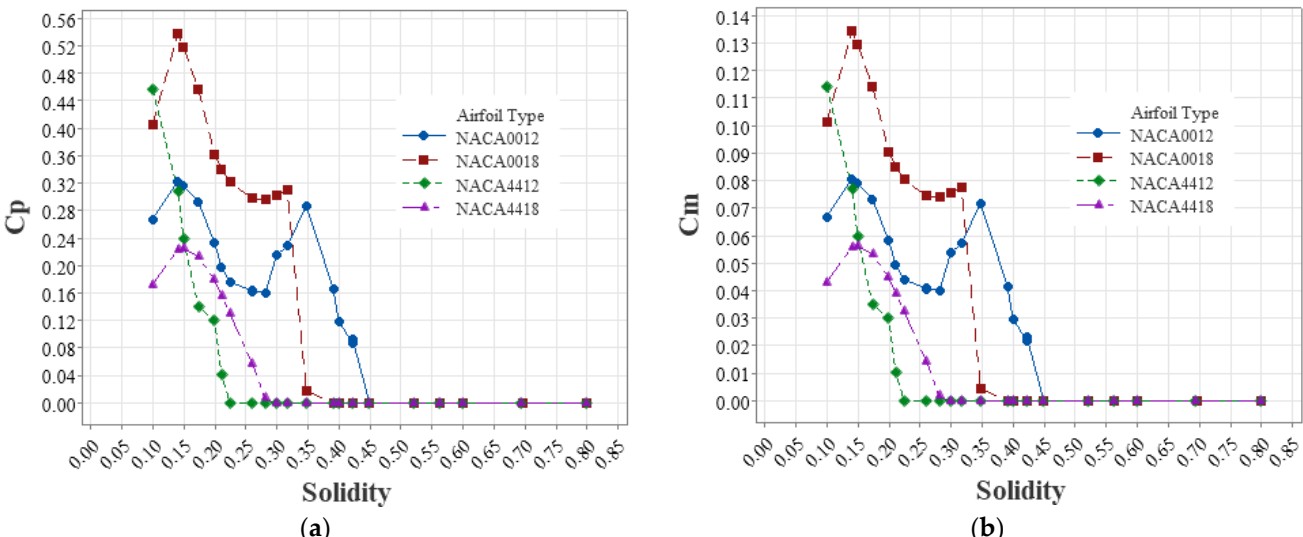

**Figure 13.** Effect of solidity on (**a**) Cp and (**b**) Cm in TSR = 4.

## 4. Conclusions

The power and momentum coefficients of various types of wind turbines were investigated in this study. The distinct and general effects of four factors, i.e., airfoil type, chord length, H/D ratio and number of blades on the performance of Darrieus wind turbine with straight blades at low Reynolds were studied. All simulations were performed using a DMST model. First, the model outcomes were validated based on the experimental results of Elkhoury et al. (2015) [24]. All of the new findings correlated well with previous experimental and modeling findings [30,31]. The power coefficient predicted in the DMST model at high TSR may not be well correlated with the experimental results [32].

In the present study, we examined the performance of the turbine at low blade speeds which this in turn increases the validity of the research results. The four factors studied are important and influential for designing wind turbines, so that after determining the working conditions of the turbine, the best situation can be selected for the proper operation of the turbine. In the following section, we outline our findings.

- Longer chords length for wind turbines can provide higher power and momentum coefficients at lower blade tip speeds, but at higher blade tip speeds this is detrimental. This is the case in all four selected airfoil types.
- At low blade speeds, more blades can solve the problems of self-rotation and low momentum; however, this is a negative factor at high blade speeds.
- Except for the NACA4412 airfoil, the maximum power coefficient of the turbine with H/D ratio of 1 is greater than in the other cases, and the maximum power coefficient was achieved with H/D ratio of 0.5 in the NACA4412 airfoil.
- Among symmetrical and asymmetrical airfoils, NACA4412 outperformed others.
- Increased blade solidity had a less positive effect on turbine performance at low blade tip speeds and a negative effect at high blade tip speeds.

**Author Contributions:** Conceptualization, P.G.; methodology, G.N.; software, P.G.; validation, A.J.; formal analysis, A.J.; investigation, P.G.; resources, M.M.; writing—original draft preparation, P.G.; and B.G.; writing—review and editing, A.J.; visualization, P.G.; supervision, G.N.; project administration, G.N.; funding acquisition, M.M. All authors have read and agreed to the published version of the manuscript.

**Funding:** This research received no external funding.

**Data Availability Statement:** All data were obtained from DMST model results.

**Conflicts of Interest:** The authors declare no conflict of interest.

# Appendix A

**Table A1.** Nomenclature.

| | |
|---|---|
| c: | blade chord length |
| *Cd*: | drag coefficient |
| *Cl*: | lift coefficient |
| Cn: | normal aerodynamic force coefficient |
| Cp: | power coefficient of the wind turbine |
| Ct: | tangential aerodynamic force coefficient |
| D: | diameter of the rotor |
| Fn: | aerodynamic tangential force |
| H: | blade length |
| N: | $N$ number of the blades |
| R: | rotor radius |
| Re: | Reynolds number |
| t: | time |
| T: | torque |
| W: | wake velocity |
| U: | axial induction factor |
| $V_\infty$: | free wind velocity |
| Ve: | equilibrium wind velocity in upstream |
| V': | downstream velocity |
| Vc | chordal velocity component |
| Vn: | normal velocity component |
| | Greek letters |
| $\alpha_1$: | angle of attack for $\lambda < 1$ |
| $\alpha$u: | angle of attack for $\lambda \geq 1$ |
| $\theta$: | azimuthal angle |
| $\ddot{\Theta}$: | rotational acceleration of rotor |
| $\lambda$: | blade tip speed ratio |
| $\rho$: | density of air |
| $\sigma$: | rotor solidity |
| $\omega$: | rotational speed of rotor |

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
