# Peer review of "Analytical Study of the Impact of Solidity, Chord Length, Number of Blades, Aspect Ratio and Airfoil Type on H-Rotor Darrieus Wind Turbine Performance at Low Reynolds Number"

_sustainability, doi:10.3390/su14052623_

Round 1

Reviewer 1 Report

Dear Authors,

below you can find my comments. please address them in the revised version of your work.

  1. The topic needs to be changed. It is not clear and cannot explain what you intended to do.
  2. Line 32 and 33, the sentence is vague. In total, you have used many complex sentences which do not show what you wanted to say.
  3. Please explain the reason to choose those NACA blades. Do you have any references that they have been used in such a type of turbine?
  4. Page 3, every variable used must be explained. There is no explanation for V prime, or V infinity. WEe can guess but they must be clarified in the context as well.
  5. Page 4, equation numbers do not have similar style. They must be revised.
  6. Figure 3, the quality of the figure is very low. Also, if you had found it in a source, please cite the source.
  7. Equation 14 does not look right. Please double check it.
  8. Figure 4, please show the error as well to see how accurate your model is. The figure quality needs to be improved.
  9. Figure 5 and 6 should be explained more. The explanation is not enough for the amount of data presented in the figures.

Author Response

Response to the reviewers

We would like to sincerely thank reviewers for their very constructive and positive comments that greatly help us improve the quality of our work. A careful consideration into their comments has been taken during the revision of our manuscript.

Below are our detailed responses to all the points raised by each reviewer. 

Reviewer #1

  1. The topic needs to be changed. It is not clear and cannot explain what you intended to do.

Response: The title of manuscript was change to “Analytical Study of the Impact of Solidity, Chord Length, Number of Blade, Aspect Ratio and Airfoil Type on H-Rotor Darrieus wind turbine performance at low Reynolds Number”

  1. Line 32 and 33, the sentence is vague. In total, you have used many complex sentences which do not show what you wanted to say.

Response: The mentioned sentence replaced with new one. (page 1 line 40-42)

  1. Please explain the reason to choose those NACA blades. Do you have any references that they have been used in such a type of turbine?

Response: The reason to choose the NACA airfoils was added and two references were cited. (page 2 line 88-89)

  1. Page 3, every variable used must be explained. There is no explanation for V prime, or V infinity. WEe can guess but they must be clarified in the context as well.

Response: Some explanation about the variable were added in the context.

  1. Page 4, equation numbers do not have similar style. They must be revised.

Response: All equation number were modified.

  1. Figure 3, the quality of the figure is very low. Also, if you had found it in a source, please cite the source.

Response: Figure 3 was replaced with new one and the source of this figure was cited.

  1. Equation 14 does not look right. Please double check it.

Response: Equation 14 was modified.

  1. Figure 4, please show the error as well to see how accurate your model is. The figure quality needs to be improved.

Response: The amount of maximum error was added in the validation section and the figure 4 was replaced with new one. (page 7 line 162-164)

  1. Figure 5 and 6 should be explained more. The explanation is not enough for the amount of data presented in the figures.

Response: Some explanation was added in the results section. (page 12 line 256-261) We hope that this modification satisfies the respectful reviewer’s consideration.

Reviewer 2 Report

Articles meets journal requirements.

Author Response

Response to the reviewers

We would like to sincerely thank reviewers for their very constructive and positive comments that greatly help us improve the quality of our work. A careful consideration into their comments has been taken during the revision of our manuscript.

Reviewer 3 Report

The authors used DMST model to determine the appropriate range of important wind turbine design parameters, and simulated the performance of 144 different turbine types with respect to chord length, number of blades, H/D ratio and airfoil type. The effects of chord length, number of blades, H/D ratio and solidity in different TSR on power and momentum coefficient of rotor with 4 airfoil types were simulated and the results were analyzed in detail. Some problems are as following:

  1. The necessity to carry out the investigation at low Reynolds is not well demonstrated. What are the biggest problems with Darrieus turbine occurred regarding low Reynolds and low blade tip speeds? More detailed related descriptions should be included in the introduction section.
  2. Does the DMST model work under all range of Reynolds? The authors declaimed that their goal is to determine the design parameters at low Reynolds numbers, and all numerical simulations were performed at 6 m.s-1 of free stream velocity and low Reynolds numbers ranging from 50,000 to 100,000. What’s the difference of DMST model at high and low Reynolds?
  3. Page 2 Line 74-75, “This review is a guide for designing Darrieus turbines at low Reynolds to overcome self-starting problems.”, since this paper is not a review, the description of “This review”is inappropriate.
  4. The clarity of figures should be enhanced. For example, the axis in Figure 2 are not clear; the words in Figure 3 are not clear.
  5. What are the parameters such as chord length, number of bladesand H/D ratio in Figure 9a and 9b? These parameters should be included in the figure caption or in the corresponding text.
  6. Some sentences are difficult to understand. Please rewrite. For example, in abstract, the sentence “Among tools used for converting wind energy was the vertical-axis wind turbine (vawt)”; Page 1 Line 36-37, “Of problems considering these turbines is that the initial torque for rotation does not reach the quorum, and also power generation at low Reynolds.”; Page 2 Line 73-74, “ so the best performance of wind turbine in various TSR defined in this study.”
  7. Some equations should be aligned.

Author Response

Response to the reviewers

We would like to sincerely thank reviewers for their very constructive and positive comments that greatly help us improve the quality of our work. A careful consideration into their comments has been taken during the revision of our manuscript.

Below are our detailed responses to all the points raised by each reviewer. 

Reviewer #3

  1. The necessity to carry out the investigation at low Reynolds is not well demonstrated. What are the biggest problems with Darrieus turbine occurred regarding low Reynolds and low blade tip speeds? More detailed related descriptions should be included in the introduction section.

Response: Some explanation was added in the introduction.

  1. Does the DMST model work under all range of Reynolds? The authors declaimed that their goal is to determine the design parameters at low Reynolds numbers, and all numerical simulations were performed at 6 m.s-1 of free stream velocity and low Reynolds numbers ranging from 50,000 to 100,000. What’s the difference of DMST model at high and low Reynolds?

Response: Accuracy of the DMST model decrease in the high Reynolds numbers and high solidities. This concept was added in the DMST model section (line 112-115)

  1. Page 2 Line 74-75, “This review is a guide for designing Darrieus turbines at low Reynolds to overcome self-starting problems.”, since this paper is not a review, the description of “This review”is inappropriate.

Response: The mentioned sentence was modified.

  1. The clarity of figures should be enhanced. For example, the axis in Figure 2 are not clear; the words in Figure 3 are not clear.

Response: All figure in the manuscript were checked and figure 3 and 4 were replaced.

  1. What are the parameters such as chord length, number of bladesand H/D ratio in Figure 9a and 9b? These parameters should be included in the figure caption or in the corresponding text.

Response: Figure 9 shows the average Cp and Cm of wind turbine for each TSR. the average of turbines with different chord length, H/D ratio and number of blade for four airfoil type were calculated. (page 9 line 228-231).

  1. Some sentences are difficult to understand. Please rewrite. For example, in abstract, the sentence “Among tools used for converting wind energy was the vertical-axis wind turbine (vawt)”; Page 1 Line 36-37, “Of problems considering these turbines is that the initial torque for rotation does not reach the quorum, and also power generation at low Reynolds.”; Page 2 Line 73-74, “ so the best performance of wind turbine in various TSR defined in this study.”

Response: Whole text was checked and mentioned sentences were rewrite. (page 1 line11-13) (page 1 line 36-37) (page 2 line 77-78)

  1. Some equations should be aligned.

Response: The equations defects were eliminated. The authors tried to do their best for reviewing the paper carefully and responding all of the comments.

Round 2

Reviewer 1 Report

Dear Authors 

although you have not addressed all my comments, I agree publishing since the current version is mature enough.

Regards

reviewer

Author Response

Thank you for your comments and the opportunity to revise our paper on ‘Analytical Study of the Impact of Solidity, Chord Length, Number of Blades, Aspect Ratio and Airfoil Type on H-Rotor Darrieus wind turbine performance at low Reynolds Number’ The suggestions offered by the reviewers have been immensely helpful.

Reviewer 3 Report

The authors have answered the questions correspondingly.

Author Response

(The authors gave the same response as above.)
